# An Integrated Method of Biomechanics Modeling for Pelvic Bone and Surrounding Soft Tissues

**DOI:** 10.3390/bioengineering10060736

**Published:** 2023-06-19

**Authors:** Wei Kou, Yefeng Liang, Zhixing Wang, Qingxi Liang, Lining Sun, Shaolong Kuang

**Affiliations:** 1Department of Mechanical and Electrical Engineering, Soochow University, Suzhou 215137, China; 2College of Health Science and Environment Engineering, Shenzhen Technology University, Shenzhen 518118, China

**Keywords:** pelvis, finite element modeling, surrounding soft tissues, surgical robot

## Abstract

The pelvis and its surrounding soft tissues create a complicated mechanical environment that greatly affects the success of fixing broken pelvic bones with surgical navigation systems and/or surgical robots. However, the modeling of the pelvic structure with the more complex surrounding soft tissues has not been considered in the current literature. The study developed an integrated finite element model of the pelvis, which includes bone and surrounding soft tissues, and verified it through experiments. Results from the experiments showed that including soft tissue in the model reduced stress and strain on the pelvis compared to when it was not included. The stress and strain distribution during pelvic loading was similar to what is typically seen in research studies and more accurate in modeling the pelvis. Additionally, the correlation with the experimental results from the predecessor’s study was strong (R^2^ = 0.9627). The results suggest that the integrated model established in this study, which includes surrounding soft tissues, can enhance the comprehension of the complex biomechanics of the pelvis and potentially advance clinical interventions and treatments for pelvic injuries.

## 1. Introduction

In the field of traumatic orthopedics, pelvic fractures are a highly serious and complex form of fracture that require great concern [1]. In the USA, there are an estimated 37 pelvic fractures per 100,000 individuals per year [2], and the incidence of pelvic fractures continues to increase [3,4]. Pelvic fractures usually require a reduction operation to promote proper healing.

Traditionally, pelvic fractures were treated with open surgery, where surgeons could directly observe soft tissue distribution and choose a reduction strategy based on experience. Open surgery is always associated with massive surrounding soft tissue incisions, long recovery time, and a high surgical complication rate [5]. In recent years, more and more patients have been choosing closed reduction and stable internal fixation to overcome these shortcomings. However, this treatment has the disadvantages of low reduction accuracy and long X-ray radiation time [6,7].

With the development of surgical robotics and surgical navigation technology, the robot-assisted fracture closed reduction system was used to enhance surgical outcomes and address issues associated with pelvic fractures. The surgery can be performed either following an intraoperative master-slave teleoperation by the surgeon’s surgical experiences [8,9] or automatically according to a preoperative reduction path planned in a 3D virtual environment [10,11]. However, regardless of the method adopted, an unreasonable reduction path and unnecessary movements may lead to excessive muscle and ligament contraction resistance force, prevent correct reduction movements, require more physical load from the surgeon, and may cause secondary injury to the pelvis and surrounding tissues [12]. Additionally, compared with other treatments, surgeons using robot-assisted fracture reduction cannot feel the force on the affected patient in real time, which can impact their ability to make accurate judgments. Due to the complexity of the pelvis, the relationships between the instruments, bones, and surrounding soft tissues are not yet fully understood. 

With the advancements in computer and simulation technologies, biomechanical modeling methods have become an effective means for addressing these challenges [13,14].

In 1972, Brekelmans et al. first applied the finite element (FE) analysis to biomechanical problems [15]. It has been widely used to solve clinical problems and has gained full acceptance. It has been applied to the study of the mechanism of damage to the human pelvis during underbody blast attacks on military vehicles [16], the effect of the structure of the area between the anus and vagina on giving birth [17], the effects of different surgical fixation strategies [18], sitting positions on patient rehabilitation during pelvic fractures [19], and the simulated stresses after implantation of different artificial prostheses in the pelvis [20,21,22,23,24]. However, the biomechanical pelvic modeling methods used in these studies cannot be used in practice because they often ignore certain muscles or ligaments, which play a significant role in the production of reduction force [5]. To solve these issues, some researchers have conducted related studies on reduction force during fracture reduction procedures. Graham et al. established a musculoskeletal model of femoral shaft fracture and found significant differences in the forces and torques of reduction in a simulation comparing four different reduction paths [25]. This implies that different reduction paths have a significant effect on the forces during reduction and need to be planned and monitored to ensure that secondary injuries do not occur. Furthermore, this presents an opportunity for robotic fracture reduction to adopt force-based planning and monitoring to improve surgical efficiency and safety. Therefore, the biomechanical analysis of the pelvis is the basis for developing appropriate reduction strategies. Li et al. implemented simulation experiments and biomechanical analysis of femoral fracture reduction with muscular tissue in a robotic environment [1]. The simulation produced results for various positions of the distal fragment relative to the proximal fragment. These results served as a basis for planning the path of robot-assisted fracture reduction, with the goal of ensuring a safe procedure. In 2017, Buschbaum et al. presented a computer-based automatic path planning method that considers muscle strength simulations using the OpenSim model from Stanford University, USA [26]. The experiments, which used five femoral SYNBONE models, demonstrated that the optimal path required the least force. However, the modeling of the pelvic structure with the inclusion of the more complex surrounding soft tissues and directly using for reduction surgery planning has not been fully explored in the literature.

In this paper, we aim to summarize the related biomechanical modeling methods of pelvic structure with soft tissue around and build an integrated biomechanical model for reduction surgery planning. The model’s reliability is verified by comparing its results to existing literature. The model will establish a mathematical foundation for analyzing the effects of instrument-tissue interactions on the body’s tissues during the reduction process. Additionally, the model will aid in minimizing forces applied during reduction and identifying an optimal path.

## 2. Materials and Methods

The pelvis is made up of pelvic bones and surrounding soft tissues, which include ligaments, joints, cartilage, muscles, and other tissues. Table 1 lists the components that should be considered in pelvic biomechanics, along with the aspects to be considered based on existing literature, i.e., thickness of cortical bone (Section 2.1.1), material properties associated with cortical and cancellous locations (Section 2.1.2), meshing (Section 2.1.3), pelvic ligaments (Section 2.2.1), pelvic joints and cartilage (Section 2.2.2), and muscle (Section 2.2.3).

### 2.1. Pelvic Bone Modeling

Anatomically, bone can be divided into two structures: cortical bone, located on the outer layer of the bone and having high strength, and cancellous bone, situated within the bone and having a low density. When modeling the pelvis with bone structures from a biomechanical research perspective, three aspects must be considered: (1) cortical bone thickness; (2) material properties associated with cortical and cancellous locations; and (3) meshing problems.

#### 2.1.1. Thickness of Cortical Bone

Cortical bone thickness varies at different sites, and there is currently no established commercial software for modeling cortical and cancellous bone separately. As a result, many researchers have used a constant value of 0.45–3 mm as cortical bone thickness [16,20,28,29,30,31]. Anderson and colleagues suggested an algorithm for calculating how thick the outer layer of bone is by measuring the distance between the polygonal surfaces that represent it and the border between the outer layer and the inner layer of bone [31]. Experiments with artificial aluminum tubes showed that the thickness of the pelvic bone ranged from 0.44 to 4.00 mm, with an average thickness of 1.41 ± 0.49 mm.

Moreover, Ramayana et al. simplified the pelvis to a solid with viscoelastic material properties and created 2 mm and 3 mm bone shell FE models [32]. They discovered that the average pelvic deformation in the 3 mm shell model was more similar to actual cadaver experiments under a load of 500 N.

#### 2.1.2. Material Properties Associated with Cortical and Cancellous Locations

Because the human pelvis is complicated, the cortical and cancellous bones have different properties in different parts. According to Leung et al. [33], cortical bone density has a greater impact on strain than trabecular density. Additionally, Rho et al. found that CT gray-scale values are linearly related to pelvic bone density when calculating Young’s modulus of the pelvic bone using the following method [34].
(1)$$\rho =\left\{\begin{array}{c} 1.9\times 10^{-3}Hu+0.105,\;Hu<816\\ 7.69\times 10^{-4}Hu+1.028,\;Hu\geq 816 \end{array}\right.$$
where ρ refers to the bone density and Hu refers to the CT scanner strength (Hu).

According to Dalstra et al. [35], the connection between bone density and Young’s modulus is as follows.
(2)$$E=2017.3\rho^{2.46}$$
where E is the Young’s modulus of bone (MPa) and ρ is the bone density ($g/{\mathrm{cm}}^{3}$).

Accurately calculating Young’s modulus from CT image gray-scale values to study the mechanical properties of specific subjects is challenging due to the significant influence of gender, age, and physical condition on human bone density. Cortical bone has been simplified to a material that is elastic and has a Young’s modulus of 17 GPa and a Poisson’s ratio of 0.3. Cancellous bone, on the other hand, has been modeled as an elastic material with a Young’s modulus of 150 MPa and Poisson’s ratio of 0.2 [16,20,27,28,36]. However, these models do not consider the differences in the quality of the patient’s bone. To address this issue, Shim et al. conducted a four-point bending test. They demonstrated it is possible to assign material properties to a smaller number of Gaussian points within larger elements instead of each element in FE models, effectively reducing the computational effort [37]. 

#### 2.1.3. Meshing

When meshing, it is crucial to balance accuracy and efficiency. For the same level of solution accuracy, hexahedral meshes are faster and more efficient than tetrahedral meshes because the latter have a higher number of cells and nodes, as well as occupying more computer memory. However, the complex pores and curvature variations in the pelvic structure pose great difficulties for hexahedral meshing. To overcome this issue, Anderson et al. showed in mesh refinement experiments that the use of three-node shell elements was almost as accurate as the use of three tetrahedral elements [31]. Therefore, shell elements can be used to replace cortical bone. In studies, tetrahedral elements are usually used to delineate cancellous bone, while triangular shell elements are used to delineate cortical bone [16,27]. 

Based on the above summary, this paper models the pelvis with bone structures in a simple, accurate, and efficient manner for calculations. To simplify calculations for the model, the average cortical bone thickness calculated by Anderson et al. was used. In addition, the material properties of the cancellous bone were obtained by acquiring Young’s modulus information from the gray-scale information–Young’s modulus relationship [31] while considering individual differences in patient constitution. To lessen mistakes brought on by artifacts from edge effects, cortical bone, which contains the surface components of the pelvis, was considered independently and given a material property of a constant Poisson’s ratio of 0.26 and a Young’s modulus of 17 GPa [33]. Finally, a combination of triangular surface mesh and tetrahedral volumetric mesh was used for mesh generation to improve calculation efficiency.

### 2.2. Modeling of Surrounding Soft Tissues

#### 2.2.1. Modeling of Pelvic Ligaments

The pelvis has a complex ligament structure primarily composed of elastic and collagen fibers. Elastic fibers allow for stretching under load, while collagen fibers provide rigidity and strength. The ligaments typically have nearly parallel fiber arrangements, which means their function is relatively specialized. They mainly connect joints and limit their movement, ensuring stability to the pelvis.

Early studies on pelvic biomechanics did not consider the influence of ligaments. However, Conza et al. demonstrated in cadaveric experiments that ligaments play a role in pelvis dynamics [39]. Later researchers combined FE and cadaveric experiments to demonstrate the importance of ligaments in the statics of the human pelvis [30,40,41,42]. To account for the effects of ligaments, Zheng et al. introduced spring elements with constant stiffness coefficients in a FE model [43]. Nonlinear spring elements were suggested by Eichenseer and Ivanov based on force-deformation models of knee ligaments [31,44,45,46], but differences between knee and pelvic ligaments may affect model accuracy. To better understand pelvic ligament properties, Cosson et al. measured their strength in cadaver experiments [47], and Hammer et al. measured their length, height, width, and cross-sectional area at the origin, midpoint, and insertion points using anatomical measurements and MRI modeling [48,49,50]. These measurements provide direction and reference for subsequent ligament research.

#### 2.2.2. Modeling of Pelvic Joints and Cartilage

The pelvic structure has several joints in the hip, sacroiliac, symphysis pubis, and sacrococcygeal areas. These joints possess a smooth layer of cartilage on their articular surfaces to reduce friction during movement and provide elasticity to absorb shocks and vibrations. In FE analysis, the sacrococcygeal joint has been considered a fixed connection to the sacrum and of little significance to stresses and strains [19,20,51]. However, researchers have shown that the sacroiliac joint is actually a movable joint despite it being originally thought to be fixed [52,53]. A limited range of motion in the sacroiliac joint can cause significant stress and strain deviations in FE analysis [28,33]. Due to its strong kinematic capability, the hip joint is typically treated as a contact with a small or frictionless coefficient [54,55]. 

The joint’s surface is covered with cartilage, which reduces friction between the bones and absorbs pressure and shock in the joint. To investigate the material properties of cartilage, Finlay and Repo et al. conducted Young’s modulus compression tests on 48 anatomical specimens of articular cartilage [56,57]. They then used the experimental data to fit a quadratic polynomial equation that provided insights into the mechanical behavior of cartilage under compression.
(3)$$E=\left\{\begin{array}{c} 0,\;\;\;\;\;\;\;\;\varepsilon \geq 0\\ 20.17\varepsilon +234\varepsilon^{2},\;\;\varepsilon <0 \end{array}\right.$$
where *E* refers to the Young’s modulus of cartilage (MPa) and *ε* refers to the cartilage strain.

To simplify the cartilage model, some researchers approximated cartilage as a linear elastic material [41,55,58], which ignores the variation of Young’s modulus with strain and introduces errors. Li et al. proposed representing cartilage as a three-parameter Mooney-Rivlin hyperplastic material based on average experimental joint compression data of the pubic symphysis. They calculated its parameter values based on gender differences to address this limitation [38]. Anderson et al. approximated cartilage as an incompressible neo-Hookean hyperplastic material and used a strengthened Lagrangian approach to ensure its incompressibility [59]. In addition, Ramezani et al. compared linear elastic, viscoelastic, and hyperplastic materials in the FE model experiment and found that viscoelastic materials were more realistic than linear-elastic materials, while hyperplastic materials had better response properties under small loads [32]. 

#### 2.2.3. Pelvic Muscle Modeling

The muscles that affect the force on the pelvis during exercise are mainly the gluteal muscle group, the piriformis muscle group, and some of the trunk muscles. The gluteal muscle group is responsible for hip extension, leg movement, and pelvic stabilization. The piriformis muscle group is located beneath the gluteal muscles and contributes to the outward rotation of the hip joint. The trunk muscles that connect to the hip bone include the iliacus, psoas, and abdominal muscles. Currently, most muscle research is based on the Hill-type model [60]. This model simplifies muscles as active contraction and passive spring units and tendons as nonlinear spring units. There is an angle α between muscles and tendons, as shown in Figure 1.

The effects of muscles are commonly considered in two methods for biomechanical analysis of the pelvis: (1) by incorporating muscle models into FE models or (2) by calculating the forces acting on each muscle at a specific instant and applying them to the FE model in the corresponding direction.

Phillips et al. created a model of the muscles in the pelvis using a type of model known as the Hill-type model [27]. In their study, they utilized spring elements to simulate the mechanical properties of the model while assuming that passive muscle stiffness was equivalent to active stiffness. To develop this model, anatomical and mechanical data from previous studies were utilized [61,62,63]. Since active muscle contraction is inhibited during robotic surgery by using muscle relaxants, the assumptions made by Phillips et al. are appropriate for the robotic surgical environment. The equivalent spring stiffness of the muscle is calculated as follows.
(4)$$k^{s}=k_{iso}^{ML}\frac{A^{S}}{A^{ML}}$$
where $k^{S}$ refers to the equivalent spring stiffness coefficient, $k_{iso}^{ML}$ refers to the isometric stiffness value, $A_{ML}$ is the area of the cortical bone to which the muscle is attached, and $A^{S}$ refers to the area of the cortical bone node where the equivalent spring element is connected.

The muscle model presented here is suitable for FE analysis and can evaluate the impact of various loads and boundary conditions on the pelvis. However, this modeling method, based on anatomical measurements, is difficult to apply for the biomechanical analysis of the pelvis in vivo or in specific objects. Therefore, Watson and Ravera utilized the AnyBody (AnyBody Technology, Aalborg, Denmark) and OpenSim (Stanford University, Stanford, CA, USA) software to compute the muscle force of a muscle-skeleton model using the Hill-type model through the use of inverse kinematics and inverse dynamics tools [28,54]. They combined the positional data collected by the optical motion capture device and the mechanical data collected by the mechanical sensors and then loaded them into the FE model.

### 2.3. An Integrated Biomechanical Modelling Method for Pelvis with Surrounding Soft Tissues

The previous sections have summarized how previous studies have modeled various pelvic components and established a theoretical foundation for the pelvis study. However, these models were often oversimplified and lacked a complete modeling strategy. An appropriate biomechanical modeling approach was proposed by extracting methods from a summary and comparing existing biomechanical studies and modeling techniques for the pelvic bone and surrounding soft tissues. This approach includes the pelvic bones, cartilage, muscles, ligaments, and other tissues and encompasses all tissue components that influence the fracture reduction forces. Each component of the pelvis was modeled as follows (see Table 2 and Figure 2):

To construct a model of the pelvic bone, the relationship between grey-scale information, bone density, and Young’s modulus proposed by Rho et al. and Dalstra et al. was used to derive a Young’s modulus of cancellous bone [34,35]. The study also considered the material properties of cortical bone, which contains surface elements of the pelvis, separately.

To simulate the pelvic ligaments, this paper employed spring elements based on the data collected by Hammer et al. [48,49,50]. 

To model the cartilage, the Mooney-Rivlin three-parameter hyperplastic model suggested by Li et al. was used [38]. This model represents the cartilage at the intervertebral disc, pubic symphysis, and acetabular.

To model the muscles, anatomical measurements of muscle stiffness results and connection areas were transformed into spring stiffness information, following the approach proposed by Phillips et al. [27]. This information was then integrated with the model proposed by Volinski et al. [58]. 

According to related research [29,34,52,53,54,55], the sacroiliac and acetabular joints are mobile joints without frictional connections, allowing for natural flexion and extension movements.

**Figure 2 bioengineering-10-00736-f002:**
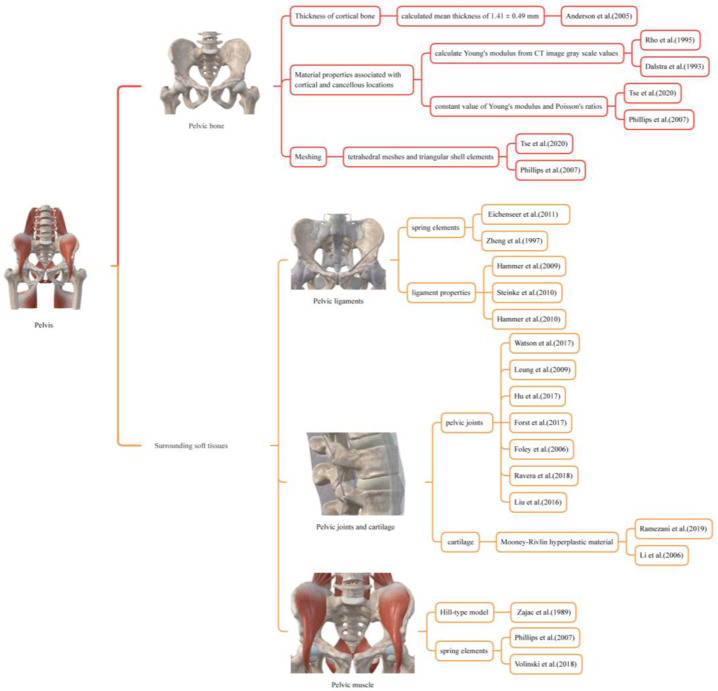
The Pelvis Modeling Method used for biomechanical modeling of the pelvic bone and surrounding soft tissues [16,27,28,31,32,33,34,35,38,43,48,49,50,51,52,53,54,55,56,57,58,60].

**Table 2 bioengineering-10-00736-t002:** Pelvis modeling methods.

Component	Material Properties	
Bone Structures	Anderson et al. [32], Rho et al. [35], Dalstra et al. [36]	
Cortical bone thickness	1.41 mm	
Young’s modulus of cortical bone	17 GPa	
Young’s modulus of cancellous bone	Equations (1) and (2)	
Poisson’s ratio	0.26	
Material	Viscoelastic materials	
Mesh	Triangular surface, Tetrahedral meshing	
Ligaments	Phillips et al. [27], Eichenseer et al. [30], Zheng et al. [43], Hammer et al. [48,49,50]	
Component	Spring Stiffness (N/mm)	No. of springs 10
Anterior sacroiliac ligaments	230	
Long posterior sacroiliac ligaments	150	6
Short posterior sacroiliac ligaments	150	4
Sacrospinous ligaments	200	3
Sacrotuberous ligaments	80	6
Iliolumbar ligaments	200	5
Inguinal ligaments	250	1
Supraspinous ligaments	15	1
Intertransverse ligaments	15	1
Muscle	Phillips et al. [28], Volinski et al. [58]	
Component	Spring Stiffness (N/mm) × Number	$A^{s}\;\left({\mathrm{mm}}^{2}\right)$
Gluteus maximus	962 × 5	4822
Gluteus medius	-	-
Gluteus minimus	-	-
pectineus	158 × 2	196
Adductor magnus	250 × 1	330
Adductor longus	67 × 2	95
Piriformis	168 × 1	60
Gemellus superior	198 × 3	667
Gemellus inferior	-	-
Cartilage	Li et al. [38]	
Materials	Mooney–Rivlin hyperplastic material C_10_ = 0.1, C_01_ = 0.45, C_11_ = 1.67, ν = 0.2	

Note: “-” symbols indicate co-modeling with the above muscles in the table.

## 3. Validation of the Finite Element Model of the Pelvis

### 3.1. Comparison Experiments of the Finite Element Model with and without Surrounding Sof Tissues

In this experiment, two FE models of the pelvis were presented to investigate how the surrounding soft tissues affect pelvic stress and strain. In Model I, no muscles or ligaments were added, and all joints were fixed to prevent deformations that would not converge in the force analysis during loading. In Model II, muscles and ligaments were included by replacing them with spring elements, and material parameters were determined using the biomechanical modeling approach proposed in this study. The sacroiliac and acetabular joints were allowed to move freely without friction, allowing flexion and extension movements. Both models were placed in identical boundary conditions to simulate a person’s natural standing position. The femur bones were fixed the vertebrae were constrained to have only vertical displacement. A vertical force was applied downwards onto the top of the lumbar spine in the model. The loading conditions were consistent between the two models, with a range of 50 N to 550 N applied in five loading steps, each with an interval of 100 N. The pelvic model comes from a healthy Chinese volunteer’s CT (female, age 43, height 162, weight 48) and then reconstructed by MIMICS 20.0 and Geomagic 2017 software. All the FE experiments were conducted in FE software ANSYS 17.0.

### 3.2. Comparison between the Finite Element Model Simulation and a Cadavic Experiment

In this experiment, we aim to evaluate our proposed integrated biomechanical FE model compared with in vitro pelvic experiment. Hao et al. conducted an in vitro pelvic experiment to obtain bone surface strains using rosette strain gauges and eliminate pelvic relaxation by applying vertical preloaded cycle forces to the pelvis [42]. This method provides accurate strain measurement and accurately reflects the strain generated when the pelvis is under load. To test the bionic reliability FE model of the pelvis, it was simulated and compared with the results of Hao’s experiment [42]. The FE model came from the same volunteer as described in Section 3.1, whose personal physical condition was very similar to the experiment subject’s in vitro pelvic experiment. The boundary conditions of the FE model were set according to Hao’s experiment [42]. The vertebrae were constrained to only have vertical displacement. The femur bones were assumed to be fixed, and there was no friction in the sacroiliac and acetabular joints. Figure 3 shows seven different locations selected as measurement points (consistent with in vitro pelvic experiment, located by experienced surgeons). To simulate the effect, six vertical forces between 50 and 550 N were applied in 100 N intervals to the top of the L4 vertebral body.

## 4. Results

Loads ranging from 50 N to 550 N in 100 N increments were applied to two FE models that used different modeling methods. As shown in Figure 4 and Figure 5, the two models had noticeably different results in stress and strain. Comparing Model I and II, it was found that the maximum stress value in Model I was 1.7 times higher than that in Model II. Additionally, the average von Mises stress values in Model II were 34.86% lower than in Model I, except for the first step of the loading experiment, where the average stress did not change significantly (less than 7.3%). In the other five steps of the loading experiments, the average stress decreased significantly in Model II, ranging from 30.2% to 34.86%. Furthermore, the maximum strain value was significantly higher (1.7 times to 1.9 times) in Model I than in Model II. Conversely, the average strain decreased significantly in Model II, ranging from 24.86% to 34.76%.

### 4.1. Model Stress Analysis

To analyze the stress–strain distribution, a 550 N vertical load was applied. Figure 6 presents the results of the Von Mises stress analysis for both Model I and Model II. The maximum stress in Model I was 11.248 MPa at the top of the first sacral joint, with a mean stress of 0.087251 MPa. The largest stresses were found in the upper part of the vertebrae, sacrum, and femur, as well as the sacroiliac joint. The stresses at the upper edge of the iliac bone and pubic joint were smaller, consistent with the results of Dolstra et al. [64]. In Model II, the highest stress still happened in the upper part of the first sacral joint, but it was reduced by 45.18% to 6.1664 MPa compared to Model I. The average stress was 0.058051 MPa, a reduction of 33.47% from Model I. Including muscles and ligaments in the model led to reductions in stress, confirming findings from earlier studies by Phillips et al. and Ravera et al. [27,54]. Figure 6 also shows that the stress distribution over the lumbar spine was significantly reduced by the supraspinous, intertransverse, and iliolumbar ligaments. Surrounding soft tissues such as the sacral tubercle ligament, sacrospinous ligament, and latissimus dorsi also helped to reduce stress concentration in the sacrum. Furthermore, the three gluteal muscles, the superior and inferior twins, the pubococcygeus, and the long retractor muscles all decrease stress on the upper part of the femur and change the distribution of stress in the pubis, ischium, and iliacus, which was consistent with the finding of Ravera et al. [54].

### 4.2. Model Strain Analysis

As seen in Figure 7, the strain analysis results for both Model I and Model II under a 550 N vertical load. The maximum strain value for Model I was 0.37597 mm/mm, which occurred at the sacroiliac joint cartilage, with an average strain of 0.0041141 mm/mm. The strain distribution is very similar to the stress distribution described above, with large strains observed in the upper part of the vertebrae, sacrum, and femur, as well as in the sacroiliac joint. These are the most common places where the pelvis can break [65]. Smaller strains were observed in the upper border of the ilium and the pubic symphysis, which aligns with the findings of Sola et al. [66]. In Model II, the maximum strain value was only 0.2219 mm/mm, a 40.98% reduction relative to Model I, which occurred in the sacrum. This is because the sacroiliac joint in this model was a mobile joint without frictional connections, and its deformation was effectively reduced by the combined action of the pelvic circumferential ligament groups [29]. Additionally, the average strain in Model II was reduced by 34.15% compared to Model I, at 2.709 × 10^–3^ mm/mm. These results suggest that the equivalent elastic strain distribution of Model II appeared to be more uniform.

### 4.3. Linear Regression Analysis

To validate the FE model, seven strain measurement points were used. These points were selected based on previous studies and the cadaver experiments conducted by Hao et al. [42]. The strain results for these points in the FE model loading experiment are listed in Table 3.

The regression equation and correlation coefficients can be determined using the cadaveric experimental data as the independent variable and the FE model loading experimental data as the dependent variable, as shown in Equation (5). Figure 8 shows the linear regression analyses and the distribution of experimental results.
(5)$$y=-0.000002x+0.923197{\;\;\;\;\;\;R}^{2}=0.9627$$
where x represents the equilibrium strain from the cadaver experiment, and y represents the equilibrium strain from the FE analysis.

The linear regression equation had a slope of 0.923197, indicating that the strain value of the FE model loading experiment was slightly lower than that of the cadaver experiment. This difference may be due to the muscles and surrounding soft tissues being removed from the surface of the pelvis in the cadaver experiment.

The strain comparison results between Model I and Model II show that the surrounding soft tissue structure effectively reduces the strain on the pelvis. However, the pelvic FE analysis data used in this study differed from the cadaver experiments regarding gender, age, and physical condition. The FE modeling included the femur and several lumbar vertebrae, whereas the lumbar vertebrae were not available in the cadaver experiments, and the femur was replaced by a femur steel implant. Therefore, the slope of 0.923197 resulted from the complex relationship of multiple factors.

The correlation coefficient R^2^ was 0.9627, proving that the pelvic cadaver experiment and the FE model loading experiment were strongly correlated and supported the modeling method proposed in this study.

## 5. Discussion

To safely reduce pelvic fractures, it is important to have a comprehensive understanding of the biomechanical response of the pelvic bone and surrounding soft tissues. FE models are valuable tools for simulating tissue stresses and strains in vivo, as instrument-tissue interaction forces change during reduction, and for predicting the mechanical response of the pelvic structures. However, several factors, such as boundary conditions, material properties, and pelvic integrity, can affect the results of FE models used in pelvic fracture reduction. Therefore, developing an integrated FE model that considers these factors is critical to improving the accuracy of fracture reduction and achieving better treatment outcomes. This paper presents an integrated biomechanical FE pelvic model that includes bone and surrounding soft tissues based on a summary and compilation of the literature.

Under human standing conditions, the results of the intact pelvis FE model indicate the effectiveness of muscles and ligaments in reducing stress concentration in the pelvic structure and altering stress and strain distribution. This is consistent with previous research by Phillips et al. and Hammer et al. [27,41]. Pelvic Model II shows a more uniform and reduced distribution of stress and strain compared to Model I. Incorporating the soft tissues surrounding the pelvis results in a more precise depiction of its movements in living individuals.

Cadaveric experiments often differ from actual biomechanical conditions of a living body because muscles, ligaments, and other surrounding soft tissues that have been rendered severely inactive on the surface of the pelvis are typically removed beforehand. Therefore, it is reasonable to expect lower strain values from the FE model loading experiment than from cadaveric experiments. The significant correlation between the FE analysis results of Model II and Hao’s cadaveric experimental results [42] confirms the model’s accuracy.

## 6. Conclusions

In conclusion, this study has successfully established an integrated FE model of the pelvis that includes bones and surrounding soft tissues. The simulation comparison with a pelvic model lacking muscles and ligaments confirms that this model more accurately represents the behavior of the pelvis in living subjects. The experimental validation results were analyzed and compared with existing strain data from the literature [42], further proving that the FE model is valid and accurate.

Moreover, the model provides surgeons with valuable information about the internal environment, which can aid in surgical decision-making and develop less invasive reduction pathways that minimize the risk of secondary injuries. During surgery, the pelvic model, combined with information from the navigation system, can calculate and monitor the force exerted on the pelvis during the reduction process in real time. This helps avoid accidental excessive reduction force that can cause injury to the patient.

The model provides important theoretical support for the development and application of robot-assisted closed pelvic reduction systems, enhancing the system’s safety and providing a possibility for further clinical application. Future research will focus on validating the model’s effectiveness and accuracy through in vitro and cadaver experiments of robot-assisted closed reduction.

## Figures and Tables

**Figure 1 bioengineering-10-00736-f001:**
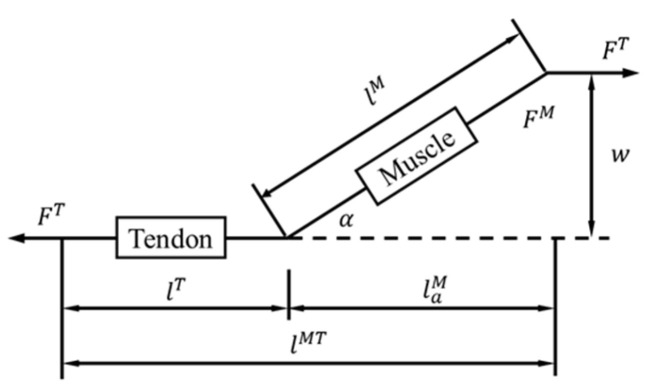
Hill-type muscle model [60].

**Figure 3 bioengineering-10-00736-f003:**
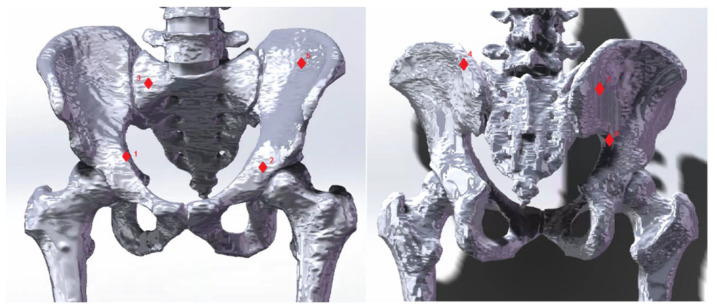
Seven strain measurement points. The seven red diamond markers indicate strain measurement points used to validate the finite element (FE) model. These markers reference cadaveric experiments conducted by Hao [42]. The numbers 1–7 correspond to the following landmarks: the midpoint of the iliopectineal line; the central point of the acetabular inner plate; a point on the 1st sacral vertebra near the sacroiliac joint; a point on the ilium of the sacroiliac joint as high as the 1st sacral vertebra; the iliac fossa; the highest point of the ischial notch; and a point on the posterior ilium near the sacroiliac joint.

**Figure 4 bioengineering-10-00736-f004:**
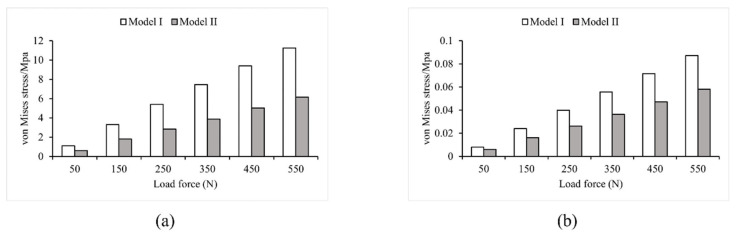
Von Mises stress values in Model I and Model II (**a**) maximum stress; (**b**) average stress.

**Figure 5 bioengineering-10-00736-f005:**
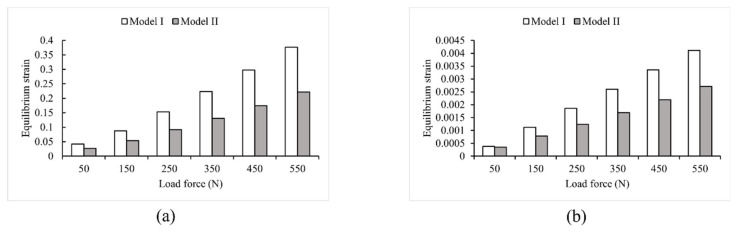
Strain values in Model I and Model II (**a**) maximum strain; (**b**) average strain.

**Figure 6 bioengineering-10-00736-f006:**
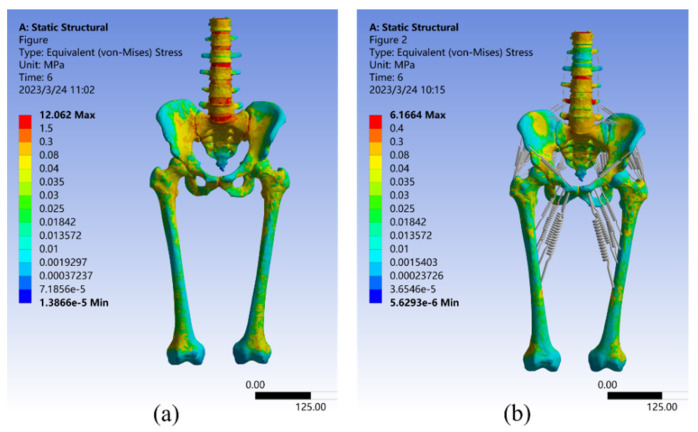
Von Mises stress distribution in Model I (**a**) and Model II (**b**) under a 550 N vertical load.

**Figure 7 bioengineering-10-00736-f007:**
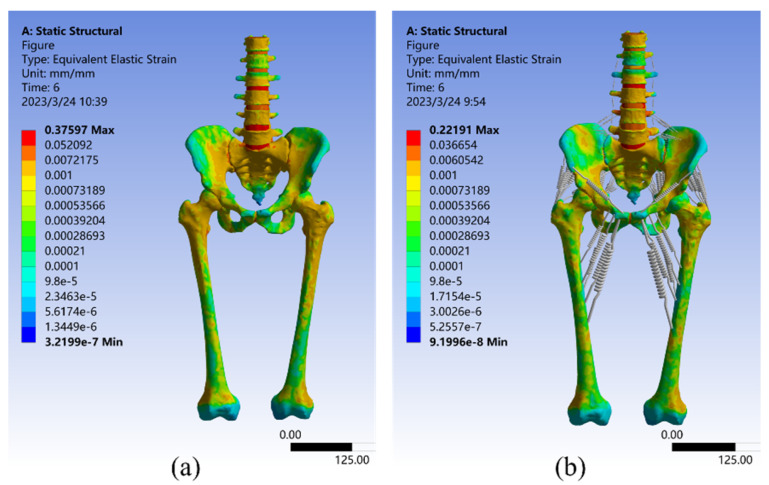
Strain distribution in Model I (**a**) and Model II (**b**) under 550 N vertical load.

**Figure 8 bioengineering-10-00736-f008:**
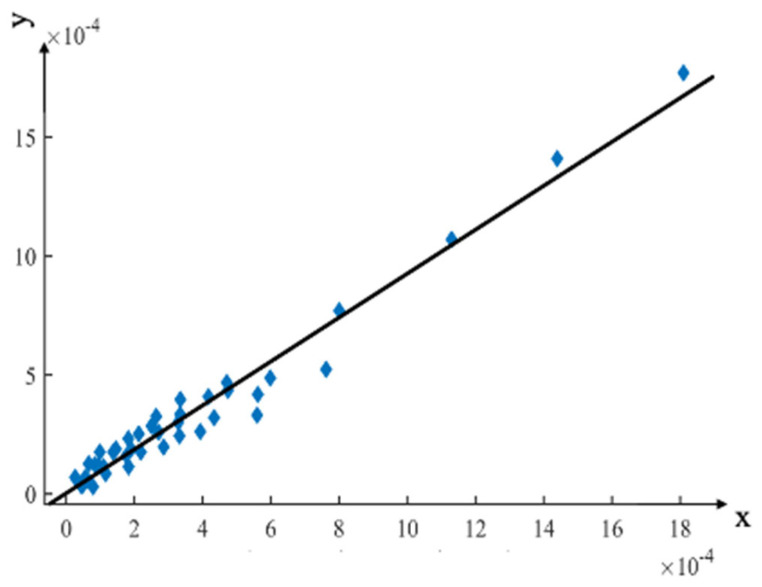
Linear regression analysis compared the FE analysis results with the cadaveric experimental results. The *x*-axis represents the equilibrium strain from the cadaver experiment, and the *y*-axis represents the equilibrium strain from the FE analysis.

**Table 1 bioengineering-10-00736-t001:** The components of the pelvis and the method of modeling.

Components	Key Aspects	Parameter Configuration	References
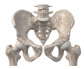 Pelvic bone	thickness of cortical bone	constant value of 0.45–3 mm	Tse et al. [16], Liu et al. [20], Phillips et al. [27], Watson et al. [28], Hammer et al. [29], Eichenseer et al. [30]
calculated mean thickness of 1.41 ± 0.49 mm	Anderson et al. [31]
2 mm and 3 mm bone shell	Ramezani et al. [32]
material properties associated with cortical and cancellous locations	calculate Young’s modulus from CT image gray-scale values	Leung et al. [33], Rho et al. [34], Dalstra et al. [35]
constant value of Young’s modulus and Poisson’s ratios	Tse et al. [16], Liu et al. [20], Phillips et al. [27], Watson et al. [28], Sichting et al. [36]
assign material properties	Shim et al. [37]
Meshing	hexahedral meshes	Li et al. [38]
tetrahedral meshes and triangular shell elements	Tse et al. [16]Anderson et al. [31], Phillips et al. [27]
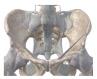 Pelvic ligaments	influence of ligaments	/	Hammer et al. [29], Conza et al. [39], Qu et al. [40], Hammer et al. [41], Hao et al. [42]
introduced spring elements	constant stiffness coefficients	Zheng et al. [43]
nonlinear spring elements	Eichenseer et al. [30], Ivanov et al. [44], Butler et al. [45], Wismans et al. [46]
ligament properties	/	Cosson et al. [47], Hammer et al. [48,49,50]
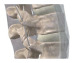 Pelvic joints and cartilage	influence of pelvic joints	/	Li et al. [19], Liu et al. [20], Watson et al. [28], Leung et al. [33], Hu et al. [51], Forst et al. [52], Foley et al. [53], Ravera et al. [54], Liu et al. [55]
influence of cartilage	material properties of cartilage	Finlay et al. [56], Repo et al. [57]
linear elastic material	Hammer et al. [41], Liu et al. [55], Volinski et al. [58]
Mooney-Rivlin hyperplastic material	Ramezani et al. [32], Li et al. [38]
neo-Hookean hyperplastic material	Ramezani et al. [32], Anderson et al. [59]
viscoelastic material	Ramezani et al. [32]
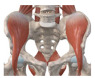 Pelvic muscle	/	Hill-type model	Zajac et al. [60]
/	incorporating muscle models into FE models	Phillips et al. [27], Delp et al. [61], Friederich et al. [62], Wickiewicz et al. [63]
/	calculating the forces and applying them to the FE model	Watson et al. [28], Ravera et al. [54]

Note: “/” symbols indicate not applicable.

**Table 3 bioengineering-10-00736-t003:** Strain results of seven points of the FE loading experiment.

	50 N (mm/mm)	150 N (mm/mm)	250 N (mm/mm)	350 N (mm/mm)	450 N (mm/mm)	550 N (mm/mm)
1	3.08 × 10^−5^	8.57 × 10^−5^	1.14 × 10^−4^	1.97 × 10^−4^	2.62 × 10^−4^	3.31 × 10^−4^
2	6.94 × 10^−5^	1.27 × 10^−4^	1.89 × 10^−4^	2.52 × 10^−4^	3.26 × 10^−4^	3.97 × 10^−4^
3	1.76 × 10^−4^	4.68 × 10^−4^	7.70 × 10^−4^	1.07 × 10^−3^	1.41 × 10^−3^	1.77 × 10^−3^
4	6.07 × 10^−5^	1.14 × 10^−4^	1.76 × 10^−4^	2.44 × 10^−4^	3.21 × 10^−4^	4.18 × 10^−4^
5	3.83 × 10^−5^	1.15 × 10^−4^	1.88 × 10^−4^	2.59 × 10^−4^	3.34 × 10^−4^	4.09 × 10^−4^
6	3.51 × 10^−5^	6.88 × 10^−5^	1.23 × 10^−4^	1.76 × 10^−4^	2.32 × 10^−4^	2.85 × 10^−4^
7	9.54 × 10^−5^	4.11 × 10^−4^	7.21 × 10^−4^	1.02 × 10^−3^	1.32 × 10^−3^	1.61 × 10^−3^

## Data Availability

Not applicable.

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
