# Peer review of "An Integrated Method of Biomechanics Modeling for Pelvic Bone and Surrounding Soft Tissues"

_bioengineering, 2023, doi:10.3390/bioengineering10060736_

Round 1

Reviewer 1 Report

The paper is an interesting survey on pelvis  biomechanical modeling, reviewing the literature. 

The paper in my opinion is ready to be printed, but it need a clarification in some points: some figures, graphs and tables are newly generated by the authors using previous data, but some seems to be taken from the literature (e.g. fig 7). So, it must ne reported in every lengend of each figures, graphs and tables the original sources.

Reviewer 2 Report

1) Do you mean to say that modelling on pelvic bone has not been attempted by any other researchers? Justify

2) The introduction part seems to be a survey. It should have been a critical analysis describing the demerits of the existing modelling methodologies.

3) Figure 1 is not clear. Make it legible.

4) Language error in line no.97

5) The first line of section 2 says a review is given in the paper. Clarify whether it is a research paper or review paper.

6) Figure 3 is unclear

7) When the focus is on pelvic bones/tissues, what is the need of a study on other tissues/bones? (section 2.2.4). How do you say that other parts affect the pelvic region/bones?

8) how did you get the results section? What is the methodology behind the experiments?

9) How do you validate that your results are the best?

Major corrections on the language is needed.

Round 2

Reviewer 1 Report

Now the sources are more clear.

Reviewer 2 Report

The article is improved. It can be accepted now.

I feel it’s fine except for minor check.